# A Novel Classification of Coronal Plane Knee Joint Instability Using Nine-Axis Inertial Measurement Units in Patients with Medial Knee Osteoarthritis

**DOI:** 10.3390/s23052797

**Published:** 2023-03-03

**Authors:** Hiroaki Tsukamoto, Kimio Saito, Hidetomo Saito, Hiroaki Kijima, Manabu Akagawa, Akira Komatsu, Takehiro Iwami, Naohisa Miyakoshi

**Affiliations:** 1Department of Orthopedic Surgery, Kita-Akita Municipal Hospital, Shimosugi, Kamishimizusawa 16-29, Kitaakita 018-4221, Japan; 2Department of Orthopedic Surgery, Graduate School of Medicine, Akita University, Hondo 1-1-1, Akita 010-8543, Japan; 3Department of Orthopedic Surgery, Omagari Kosei Medical Center, Omagari Torimachi 8-65, Senboku 014-0027, Japan; 4National Institute of Technology, Sendai College, Natori 981-1239, Japan; 5Department of System Design Engineering, Faculty of Engineering Science, Graduate School of Engineering Science, Akita University, Akita 010-8502, Japan

**Keywords:** knee osteoarthritis, inertial motion sensor units, varus thrust, substantial knee instability

## Abstract

The purpose of this study was to propose a novel classification of varus thrust based on gait analysis with inertial motion sensor units (IMUs) in patients with medial knee osteoarthritis (MKOA). We investigated thigh and shank acceleration using a nine-axis IMU in 69 knees with MKOA and 24 (control) knees. We classified varus thrust into four phenotypes according to the relative medial–lateral acceleration vector patterns of the thigh and shank segments: pattern A (thigh medial, shank medial), pattern B (medial, lateral), pattern C (lateral, medial), and pattern D (lateral, lateral). Quantitative varus thrust was calculated using an extended Kalman filter-based algorithm. We compared the differences between our proposed IMU classification and the Kellgren–Lawrence (KL) grades for quantitative varus thrust and visible varus thrust. Most of the varus thrust was not visually perceptible in early-stage OA. In advanced MKOA, increased proportions of patterns C and D with lateral thigh acceleration were observed. Quantitative varus thrust was significantly increased stepwise from patterns A to D. This novel IMU classification has better clinical utility due to its ability to detect subtle kinematic changes that cannot be captured with conventional motion analysis even in the early stage of MKOA.

## 1. Introduction

Medial compartmental knee osteoarthritis (MKOA) is the most common degenerative joint disorder. It involves cartilage wear, osteosclerosis, the formation of osteophytes, meniscal degeneration, and the development of bone marrow lesions. Various changes, such as inflammation, injury, obesity, aging, malalignment, instability, and muscle activation patterns, can induce a cartilage metabolism imbalance [1]. Among these phenomena, biomechanical factors are the main cause of osteoarthritis progression [2]. Overloading on the medial compartment due to coronal plane deformity or joint instability is a known risk factor for osteoarthritis progression.

During the stance phase of gait, the ground reaction forces (GRF) pass medially to the knee joint center, and the medial compartment bears the greatest proportion of the load [3,4]. This is exacerbated by varus knee deformity, where patients are prone to the development of more severe OA [5,6]. Varus thrust is a very visible gait abnormality that is explained as the sudden deterioration of knee alignment laterally during the stance phase. It is thought to be part of the abnormal kinematics that occurs in MKOA in the coronal plane. It is an indicator of coronal plane joint instability and an independent risk factor for MKOA that correlates with the Kellgren–Lawrence (KL) grade and clinical symptoms [7,8,9]. Varus thrust gait is caused by cartilage or meniscus loss as a result of medial compartment arthritis, relaxation of the lateral collateral ligament, and the dysfunction of muscle coordination involving “pseudolaxity” of the medial compartment, causing it to collapse at foot strike [10]. However, biomechanically, little is known about knee joint instability regarding the frontal plane in early MKOA. The contact location and shifting of load bearing regions during gait are related to the initiation of knee OA [11].

Gait analysis is an effective way to elucidate the pathophysiology of knee OA. It is reported that kinematic changes during walking affect the onset and progression of knee OA [7,12]. The occurrence of varus thrust in early MKOA coincides with much smaller differences in the joint angle that are often difficult to observe clinically. Multi-camera-based stereophotogrammetric systems combined with ground reaction force plates were used to measure the three-dimensional (3D) positions of body markers attached to the affected knee [13]. These approaches calculate the joint angle and the external knee adduction moment (EKAM) that represent a varus torque on the knee joint. The EKAM is the moment created about the knee joint center from the GRF and inertial forces. The EKAM is equal and opposite to the internal knee abduction moment, which is primarily generated by muscle forces, soft tissue forces, and contact forces. The EKAM is a surrogate marker of joint loading and is closely associated with not only radiographic findings and clinical symptoms but also varus thrust [9,14,15]. Although the EKAM can be measured quite accurately in the laboratory and is a useful method for analyzing and understanding gait, this method is hard to apply in clinical practice due to the costs associated with its expensive measurement environment and the notable effort required to collect a large amount of data for testing epidemiological hypotheses. Furthermore, gait analysis with a motion capture system has limitations in terms of capturing subtle joint instability features because the movements involved are imperceptible.

Inertial motion sensor units (IMUs), which directly measure kinetic parameters such as segment accelerations and angular velocities, are alternative devices that are widely used, inexpensive, and easy to employ clinically. Acceleration data, which can directly measure the impact transmitted to the location of the sensor, can provide a good indicator of varus thrust. This is attributable to the fact that the knee is subjected to greater forces and moves in the lateral direction during the beginning of the stance phase in patients with varus thrust, resulting in greater acceleration. Several researchers quantified varus thrust using shank acceleration data from IMUs [16,17,18,19]. However, little has been discovered about the relative femoro-tibial joint kinetic change at the initial contact through the analysis of the lateral acceleration vector in the thigh and shank. Because acceleration can also be regarded as being equivalent to force according to Newton’s second law, it is possible to capture the deviation of inertial forces that occurs early in knee osteoarthritis [20]. Thus, we would be able to obtain new insights into the subtle joint instability associated with MKOA. We hypothesized that the patterns of the acceleration vectors of the thigh and shank in the coronal plane could be related to varus thrust. The purpose of the current investigation was, primarily, to propose a novel IMU classification based on the acceleration vector patterns in the thigh and shank and, secondly, to investigate the prevalence and the degree of varus thrust for each IMU classification group and radiographic grade.

## 2. Materials and Methods

### 2.1. Participants

This cross-sectional study was approved by the ethics committee of our institution and was performed in accordance with the ethical standards of the Akita University Ethics Committee (No 2017-1775). We recruited outpatients with MKOA from our hospital between May 2017 and March 2018. All patients provided written informed consent prior to their inclusion in this study. The inclusion criteria were an age above 40 years, a body mass index (BMI) < 31 kg/m^2^, available radiographs of weight-bearing standing in a long leg anterior–posterior view, and the ability to walk without gait aids. The exclusion criteria included missing data, valgus alignment, infection, rheumatoid arthritis, prior knee surgery or knee trauma, the presence of hip and/or ankle OA on the radiography results, or the occurrence of neurological disorders or cardiovascular or peripheral artery diseases. Asymptomatic control subjects were staff and community volunteers. The inclusion criteria were no history of knee pain for the past three months, no tenderness in the medial joint space, and no radiological OA findings. The exclusion criteria were the same as for the patients.

### 2.2. Gait Analysis

For the gait analysis, we used four nine-axis inertial motion sensor units with a three-axis accelerometer, gyroscope, and magnetometer (IMU-Z2, ZMP Inc., Tokyo, Japan). Each sensor was attached to the anterior sides of the thigh and shank and secured with a flexible elastic Velcro band to compress the sensor and prevent it from shifting (Figure 1). The measurement sampling rate was 100 Hz. The participants walked at a self-selected, comfortable speed along a 10 m walkway. The present gait analysis was not only conducted for research but for clinical application. Consequently, the gait speed was not standardized, but rather, a comfortable gait speed was used to reproduce the free gait of each individual. Additionally, the subjects walked on a wooden floor wearing shoes worn daily. To stabilize each subject’s gait, spaces of at least three steps in length were set up in front and behind the 10 m walkway. The participants wearing IMUs underwent three gait tests. The average values extracted from the three gait tests were calculated to represent one data point per subject and used as the gait variable in accordance with previous studies. The subjects walked at a self-selected, comfortable speed several times while wearing the sensors with a brief measurement orientation to familiarize themselves with the walking movement. In addition, to minimize the effect of fatigue on the results of the gait test, a standing rest period of approximately ten seconds was allowed between each of the three gait tests [21,22]. One experienced expert orthopedic surgeon assessed the presence of clearly visible varus thrust. At first, the participant stood upright for 5 s, and the IMU’s initial posture was measured. The examiner then told the participant when to start and stop walking. During these gait tests, the examiner counted the participant’s steps and recorded the contact of each leg with the ground (Figure 2). All sensors were logged in sync. The obtained data were transmitted via Bluetooth to a laptop (ThinkPad X270; Lenovo Corporation, Beijing, China). Furthermore, an extended Kalman filter (EKF)-based algorithm was used to estimate the orientation of the knee [23,24,25]. All data were analyzed with MATLAB R2016a (MathWorks Inc., Natick, MA, USA). We defined quantitative varus thrust as the maximum varus angle with the largest peak knee angular velocity during the first 30% of the stance phase in accordance with a previous study [20]. The no varus thrust condition was defined as a knee without the largest peak angular velocity.

Figure 1 displays the measurement set-up for the inertial motion sensor units (IMUs) with a three-axis accelerometer, gyroscope, and magnetometer. The IMUs were attached in front of both the thigh and shank using flexible bands.

The participants wearing IMUs underwent three gait tests. To stabilize the subject’s gait, spaces of at least three steps in length were set up in front and behind the 10 m walkway. At first, the participant stood upright for 5 s to estimate the IMU’s initial posture. The examiner then told the participant when to start and stop walking. During these gait tests, the examiner counted the participant’s steps and recorded the contact of each foot with the ground.

### 2.3. IMU Classifications of the Relative Joint Behavior across the Femoro-Tibial Joint

We classified the relative acceleration vector patterns observed across the FT joint on the coronal plane at the point of initial contact into four phenotypes based on the results of the gait analysis using nine-axis IMUs: pattern A (thigh medial, shank medial), pattern B (thigh medial, shank lateral), pattern C (thigh lateral, shank medial), and pattern D (thigh lateral, shank lateral) (Figure 3).

Pattern A: The first directions of medial–lateral acceleration on the thigh and shank are medial and medial.Pattern B: The first directions of medial–lateral acceleration on the thigh and shank are medial and lateral.Pattern C: The first directions of medial–lateral acceleration on the thigh and shank are lateral and medial.Pattern D: The first directions of medial–lateral acceleration on the thigh and shank are lateral and lateral.

### 2.4. Radiographic Measurements

Radiographic grading of medial knee OA was evaluated using standing anterior–posterior knee radiographs and KL grades [26]. The scoring system was as follows: 0 (normal), 1 (possible osteophyte), 2 (definite osteophyte, possible joint space narrowing), 3 (moderate osteophyte, definite joint space narrowing, osteosclerosis), and 4 (large osteophyte, osteosclerosis, full-thickness cartilage loss).

### 2.5. Statistical Analysis

Cross tabulation of the KL grades and IMU classifications was performed. A chi-square test of independence with Fisher’s direct methods was conducted to compare the groups. The quantitative varus thrust values and gait speed were compared with an analysis of variance (ANOVA). All data were parametric and were analyzed using Tukey’s highest significant difference (HSD) post-hoc test. All data values are expressed as the mean ± standard deviation (SD). SPSS for Windows version 17.0 software (SPSS, Chicago, IL) was used for all statistical analyses. Significant differences between groups were assigned at *p* values of 0.05 or less.

A prior power analysis was conducted using G*Power version 3.1.9.5 software (University of Kiel, Kiel, Germany). Our study had a sufficient sample size (effect size, 0.4; αerror, 0.05; power (1-β), 0.8; number of groups, 4; sample size, 76).

## 3. Results

### 3.1. IMU Classification and Demographics

A flow chart for the present study is presented in Figure 4. Sixty-nine patients with medial knee OA and 24 asymptomatic controls were included in the final analysis. The distribution of KL grades was as follows: grade 0, 24 knees; grade 1, 8 knees; grade 2, 14 knees; grade 3, 31 knees; and grade 4, 16 knees. The asymptomatic controls showed no radiographic changes. In terms of the IMU classification among the study participants, 27 were classified as pattern A, 21 as pattern B, 25 as pattern C, and 20 as pattern D (Table 1). The stratification of the patients by IMU classification also revealed significant differences in age (*p* = 0.016). The cross tabulation of the KL grades and IMU classifications is presented (χ^2^ = 30.96, *df* = 12, *p* = 0.002). The proportion of KL 0–1 in pattern A was 55%, the proportion of KL 2 and 3 in patterns B and C was 38% and 64%, respectively, and the proportion of KL 4 in pattern D was 50%.

### 3.2. Gait Speed

The average gait speed decreased in a stepwise manner as the KL grade progressed. Among the IMU gait classifications, the gait speed of pattern A was significantly greater than that of the other patterns. The gait speed of pattern D was significantly slower than those of patterns B and C (*p* < 0.05, Table 2).

### 3.3. The Incidence of Clearly Visible Varus Thrust and Quantitative Varus Thrust

The relationships among the KL grade, IMU classification, and the proportion of knees with varus thrust are tabulated in Table 3. No knees with varus thrust were observed in the pattern A group in the IMU classification. Although a small number of knees revealed varus thrust in the pattern B group, the number of knees with varus thrust was greater in the pattern C group (52%) and pattern D group (90%) (Table 3). Quantitative varus thrust in the KL 3 and 4 knees measured with IMU was significantly larger than that in the control group and KL 1 and 2 participants (*p* < 0.05, Table 4 and Figure 5). There were no significant differences in the value of the quantitative varus thrust between the KL 1, 2, and control subjects. In contrast, the IMU classifications revealed significant differences between each of the groups (*p* < 0.05, Table 4 and Figure 5).

## 4. Discussion

Here, we proposed the relative acceleration patterns of the femur and shank across the FT joint using IMUs at the initial contact phase during gait. According to the current investigation, the medial–lateral acceleration vector of the thigh and shank can be classified into one of four patterns. We found a relationship between the acceleration vector pattern and the MKOA stage, even though most varus thrust cases are not visually perceptible in early-stage OA (KL < 2). In advanced MKOA, increased proportions of patterns C and D with lateral thigh acceleration were observed. Quantitative varus thrust was significantly increased stepwise from patterns A to D. We found that these novel IMU classifications have clinical utility due to their ability to detect subtle joint instability even in early-stage OA, which cannot be captured with conventional motion analysis.

There are numerous studies on coronal plane joint instability in MKOA [7]. Previous gait analyses indicate that varus thrust increases the EKAM and varus angular velocity in the early stance phase [13,14,17,27]. Our previous study demonstrated a high sensitivity for the quantitative assessment of varus thrust using the peak knee varus angular velocity obtained from nine-axis IMUs [20]. We learned through our research that subtle joint instability is detectable by measuring the knee angular velocity via IMUs, which was not possible to observe previously. This means that more patients with potential and substantial varus thrust could now be observed.

Previous investigations on the sensor-based knee dynamic instability assessment consistently mention only the amount of shank acceleration [18]. Some studies investigated the acceleration vectors in both the thigh and shank [28]; however, the relationship between the directions of acceleration in the thigh and shank across the FT joint was not mentioned. By considering the directions of the acceleration vectors obtained from the thigh and shank, the inertial force applied to the knee joint during gait can be estimated. Yoshimura et al. reported that, in 30 out of 40 asymptomatic knees, an acceleration peak in the lateral direction occurred after heel strike followed by an acceleration in the medial direction [19]. These findings are consistent with those obtained in the current study. Eighty-eight percent (21 of 24 knees) of the controls were shown to have patterns A or B where the direction of acceleration in the femur is inward. Fifty percent (12 of 24) of the controls were shown to have pattern A where the direction of acceleration in both the femur and shank is inward.

Theoretically, normal gait is a state in which an efficient inverted pendulum motion is formed, and the GRF are properly controlled during the stance phase. When a person views from the coronal plane, the foot must be placed near to the line of the center of gravity (COG), the loaded hip is in adduction, and the tibia is not vertical in the coronal plane but is also adducted. These promote a shift of the COG to the loaded limb, which aids in knee joint leveling. Normal gait provides a dynamic stability effect that induces a small shearing force to the knee. Shifts in the COG generate inertial forces in the coronal plane. The knee is more loaded in the medial compartment by an EKAM during the stance phase. However, an external knee abduction moment is generated at heel strike by inertial forces, which is sufficient to temporarily unload the lateral compartment with all the compressive forces passing through the medial compartment [3,29,30,31]. Thus, pattern A in the IMU classification is considered to reflect an optimal load distribution in the medial compartment.

It is well documented that gait speed decreases as OA progresses [32,33]. In previous studies, the acceleration detected with the IMU was correlated with the gait speed. In the current investigation, although we measured the acceleration generated at initial contact using the IMU without a fixed gait speed but rather at a self-selected, comfortable speed, the gait speed decreased as the KL grade progressed, a finding that is similar to previous reports (Table 2) [32,33]. The IMU classification identified in the current study is theoretically completely unaffected by the gait speed. Interestingly, the amount of varus thrust, which was determined even though gait speed was not standardized, was positively associated with the progression of the KL classification and IMU classification. Our results indicate that this classification would be very useful for future clinical practice.

In our current analyses, in patterns B and C, the acceleration vector of the thigh and shank is opposite in direction at heel strike. It is speculated that one of the causes is the displacement of the contact areas and increased shear stress in the knee joint [34]. Andriacchi et al. suggested that the contact location and load bearing region shifts during gait are related to the initiation of knee OA [11]. The shear stress due to recurrent non-physiological loading of the joint induce chondrocyte damage and lead to chondrocyte apoptosis [35]. Because classical tribology dictates that the amount of wear is proportional to the sliding velocity, both patterns B and C should produce large shear forces and lead to cartilage damage. We termed these acceleration mismatch patterns “substantial varus thrust”. This reflects an invisible instability of the knee joint due to a failure, such as a dysfunctional proprioception, and muscle activation patterns [9,36,37,38,39]. Muscle activities in patients with MKOA revealed an altered muscle activity pattern that unloads the medial compartment of the knee [27,29,38]. Half of early MKOA cases without varus thrust in our current study showed these substantial thrust patterns. Substantial varus thrust was shown to be a risk factor for osteoarthritis progression in the future. A lack of medial shank acceleration at the point of initial contact can even occur in asymptomatic individuals without radiological MKOA [40]. One possibility is that the tibia is directly affected by the ascending kinetic chain from the foot on the floor and thus reflects foot and ankle characteristics [41]. On the other hand, a lack of medial acceleration in the thigh at heel strike is considered a failure in which the gait strategy along with the medial translation of the center of gravity breaks down because the femur is the bone that connects the lumbar pelvic girdle to the knee joint. Further studies should investigate the relationship between the IMU classification and the full-body kinematics and kinetics.

Varus thrust was observed more in pattern D. Pattern D has an acceleration pattern in the lateral direction for both the thigh and shank. In both segments, these forces represent varus thrust gait, which is defined as an excessive “bowing out” knee motion in the coronal plane during gait that produces EKAM in the knee and a large degree of mechanical stress in the medial compartment [7,31].

We are aware of several limitations in this study. In the first instance, this was a cross-sectional retrospective study, and the number of subjects was small. However, the sample size was statistically sufficient for the power analysis. A future longitudinal investigation is needed to identify the nature of MKOA according to the IMU classification. Second, there were larger errors when estimating the joint angle using IMUs compared with other types of 3D gait analysis due to skin motion error and drift error. There is a limit to the accuracy of quantitative varus thrust estimation. However, due to recent advances in wireless capabilities and data storage in wearable technology, IMUs have become more affordable and practical for use in gait assessments. IMUs can act as a more accessible alternative to marker-based, three-dimensional motion analysis in a clinical setting. It is expected that sensors and estimation methods will improve in the future. Third, having a nonconstant gait speed depends on the magnitude of the acceleration. Gait speed is one of the critical issues associated with gait analysis using IMUs. However, the direction of acceleration was not found to depend on the gait speed. They were in different dimensions.

## 5. Conclusions

A novel IMU classification was described in the current study. The patterns of direction of the acceleration vectors of the thigh and shank in the coronal plane were classified into patterns A to D. The IMU classification was correlated with the OA stage. This novel IMU classification method has better clinical utility due to its ability to detect cases of subtle joint instability even in early-stage MKOA.

## Figures and Tables

**Figure 1 sensors-23-02797-f001:**
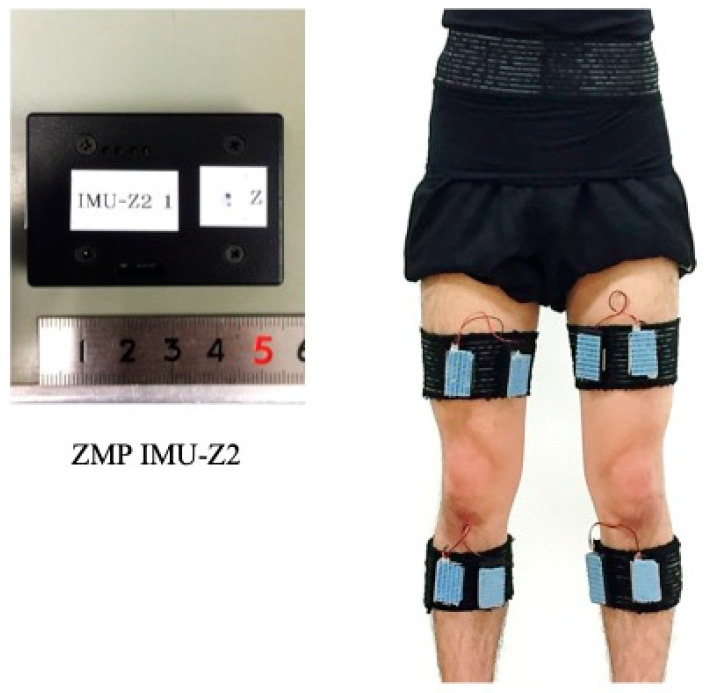
Measurement set-up.

**Figure 2 sensors-23-02797-f002:**
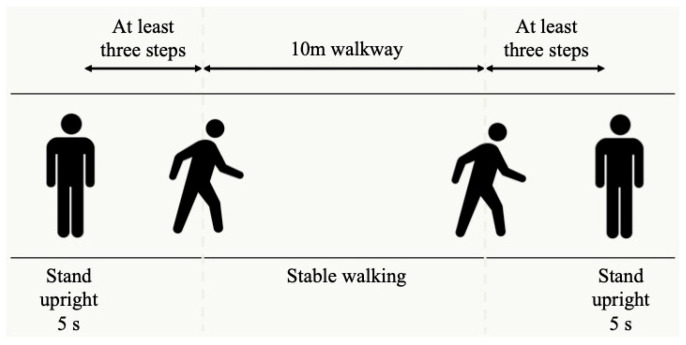
Gait analysis overview.

**Figure 3 sensors-23-02797-f003:**
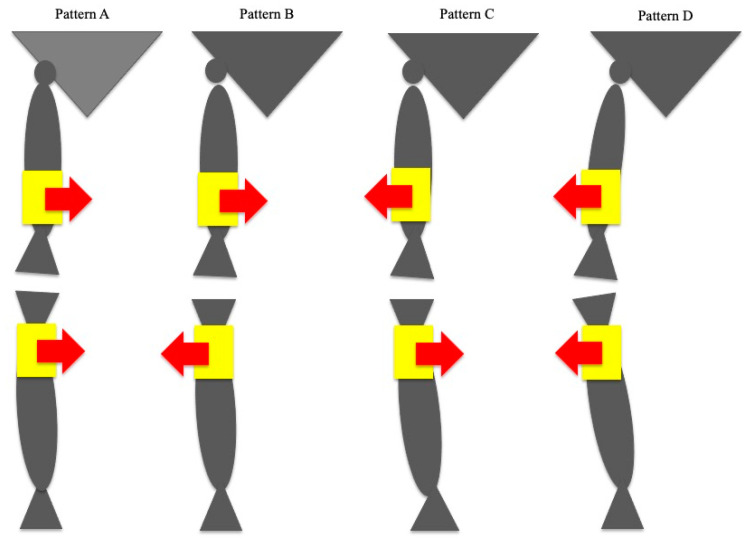
IMU classification according to the angular acceleration pattern. Red arrows: the directions of medial-lateral acceleration.

**Figure 4 sensors-23-02797-f004:**
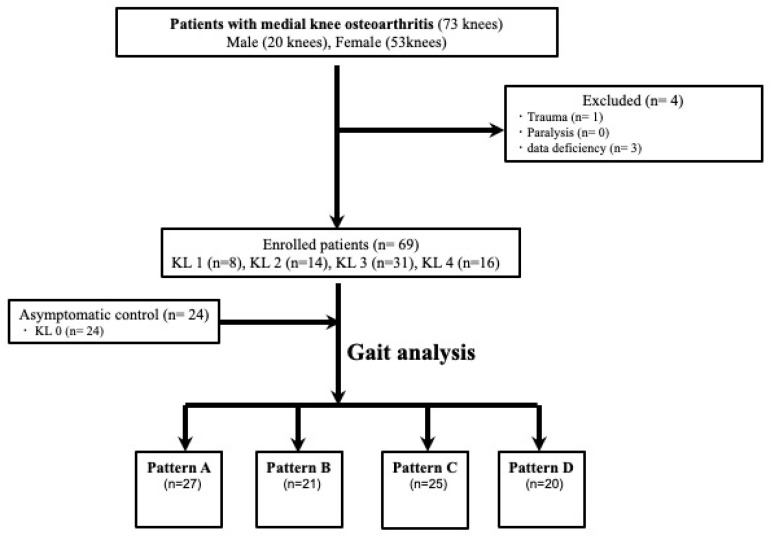
Flow chart showing the enrolled subject distribution.

**Figure 5 sensors-23-02797-f005:**
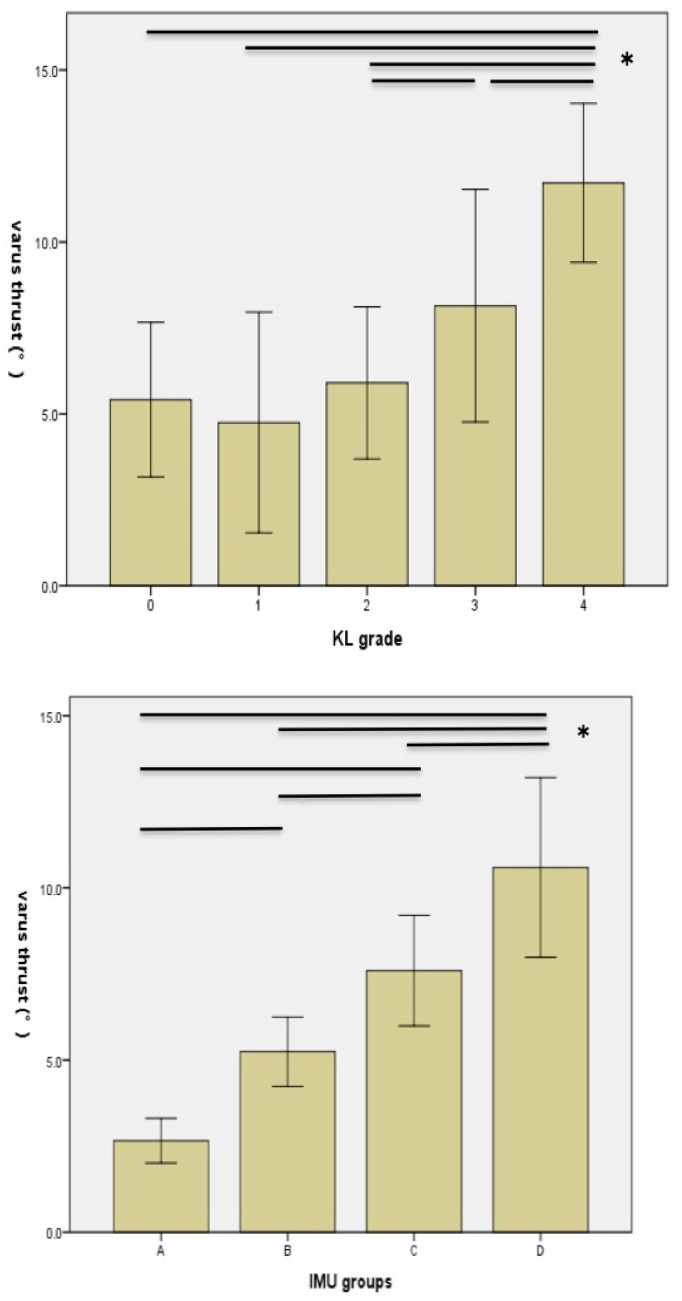
Post-hoc analysis of quantitative varus thrust by KL grade and IMU classification. * *p* < 0.05 determined with one-way ANOVA using Tukey’s HSD post-hoc test.

**Table 1 sensors-23-02797-t001:** Demographic data and measured variables for each clinical phenotype.

	Pattern A (n = 27)	Pattern B (n = 21)	Pattern C (n = 25)	Pattern D (n = 20)	*p* Value
Age	35.1 ± 17.5	40.8 ± 24.5	52.3 ± 18.8	59.3 ± 16.2	0.016
Height (cm)	158.4 ± 9.9	149.8 ± 6.2	160.9 ± 12.2	161.6 ± 7.3	0.158
Body weight (kg)	57.4 ± 11.0	69.3 ± 20.9	71.3 ± 25.9	66.9 ± 15.5	0.474
KL 0 (CTL) (n = 24)	12 (44%)	9 (42%)	3 (12%)	0 (0%)	
KL 1 (n = 8)	3 (11%)	2 (10%)	2 (8%)	1 (5%)	
KL 2 (n = 14)	3 (11%)	2 (10%)	8 (32%)	1 (5%)	
KL 3 (n = 31)	9 (34%)	6 (28%)	8 (32%)	8 (40%)	
KL4 (n = 16)	0 (0%)	2 (10%)	4 (16%)	10 (50%)	
HKA angle (°)	1.4 (3.5)	1.7 (3.9)	2.1 (3.6)	2.5 (4.3)	0.088

Values are expressed as the number of patients or the mean ± standard deviation (SD). *p* < 0.05. CTL: control, HKA: hip-knee-ankle, KL: Kellgren–Lawrence.

**Table 2 sensors-23-02797-t002:** Gait speed analysis.

	KL 0 (CTR)	KL 1	KL 2	KL 3	KL 4	Average +/− SD
Pattern A (n= 27)	1.44 (12)	1.45 (3)	1.36 (3)	1.31 (9)	N/A	1.39 ± 0.07
Pattern B (n = 21)	1.42 (9)	1.34 (2)	1.34 (2)	1.29 (6)	1.23 (2)	1.32 ± 0.11 ^‖^
Pattern C (n = 25)	1.40 (3)	1.34 (2)	1.32 (8)	1.27 (8)	1.22 (4)	1.31 ± 0.10 ^‖^
Pattern D (n = 20)	N/A	1.42 (1)	1.29 (1)	1.26 (8)	1.18 (10)	1.28 ± 0.14 ^‖¶^**
Average ± SD	1.42 ± 0.02	1.38 ± 0.09 ^§^	1.32 ± 0.07 *^†§^	1.28 ± 0.06 *^†§^	1.21 ± 0.04 *^†‡§^	

The values are expressed as the average +/− standard deviation. The number of cases in each group is given in parentheses. CTR: control group, KL: Kellgren–Lawrence, SD: standard deviation, HSD: highest significant difference; * *p* < 0.05 vs. control group with Tukey’s HSD post-hoc test; ^†^
*p* < 0.05 vs. KL 1 group with Tukey’s HSD post-hoc test; ^‡^
*p* < 0.05 vs. KL 2 group with Tukey’s HSD post-hoc test; ^§^
*p* < 0.05 vs. KL 3 group with Tukey’s HSD post-hoc test; ^‖^
*p* < 0.05 vs. pattern A with Tukey’s HSD post-hoc test; ^¶^
*p* < 0.05 vs. pattern B with Tukey’s HSD post-hoc test; ** *p* < 0.05 vs. pattern C with Tukey’s HSD post-hoc test.

**Table 3 sensors-23-02797-t003:** Proportion of knees with clearly visible varus thrust stratified by KL grade and IMU classification.

	KL 0 (CTR)	KL 1	KL 2	KL 3	KL 4	Total
Pattern A (n = 27)	0/12(0%)	0/3 (0%)	0/3 (0%)	0/9 (0%)	0/0 (0%)	0/27 (0%)
Pattern B (n = 21)	0/9 (0%)	0/2 (0%)	0/2 (0%)	2/6 (33%)	1/2(50%)	3/21 (14%)
Pattern C (n = 25)	0/3 (0%)	1/2 (50%)	4/8(50%)	5 /8(62%)	3/4 (75%)	13/25 (52%)
Pattern D (n = 20)	0/0 (0%)	1/1 (100%)	1/1 (100%)	7/8 (88%)	9/10 (90%)	18/20 (90%)
Total	0/24 (0%)	2/8 (12%)	5/14 (35%)	14/31 (54%)	13/16 (81%)	34/93 (40%)

The values are expressed as the number of knees with clearly visible varus thrust in each category (proportion). CTR: control group, KL: Kellgren–Lawrence.

**Table 4 sensors-23-02797-t004:** Quantitative varus thrust for each KL grade and IMU classification.

	KL 0 (CTR)	KL 1	KL 2	KL 3	KL 4	*p* Value
varus thrust (°)	5.4 ± 2.2	4.8 ± 3.2	5.9 ± 2.2	8.1 ± 3.4	11.7 ± 2.3	0.000
	(4.5–6.3)	(2.1–7.4)	(4.6–7.2)	(6.7–9.6)	(10.2–13.3)	
	**Pattern A**	**Pattern B**	**Pattern C**	**Pattern D**	***p* Value**	
varus thrust (°)	5.4 ± 2.2	4.8 ± 3.2	5.9 ± 2.2	8.1 ± 3.4	0.000	
	(2.3–3.0)	(4.8–5.7)	(6.7–8.5)	(9.6–11.6)		

Values are expressed as the average ± standard deviation (95% CI). CTR: control group, KL: Kellgren–Lawrence.

## Data Availability

Our data are unavailable due to privacy and ethical restrictions.

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
