# Peer review of "A Novel Classification of Coronal Plane Knee Joint Instability Using Nine-Axis Inertial Measurement Units in Patients with Medial Knee Osteoarthritis"

_sensors, 2023, doi:10.3390/s23052797_

Round 1

Reviewer 1 Report

While a simple study, the current manuscript has practical application within the healthcare community. Overall, the authors are encouraged to have an additional reviewer go over the manuscript for grammatical errors as that issue is quite extensive throughout the paper. Although there are several limitations to the study, the authors address those limitations within the discussion section so I applaud their proactive approach to recognizing potential issues, and with this honest recognition of limitations, I believe this manuscript with revisions will do well as a publication within this journal.

Within the introduction, lines 77-81 consist of a run-on sentence and lines 81-82 consist of an incomplete sentence. Rewrite both accordingly, but overall, the introduction is a good start to the manuscript. 

Within the methods section, the authors are commended for including inclusion and exclusion criteria for the MKOA sample population, but what was the inclusion and exclusion criteria for the controls? The controls throughout the manuscript are defined as having a "healthy knee", but what defines "healthy" as it relates to the knee? Authors should be cautious as the controls are also referred to as "healthy volunteers" within the discussion, but not being classified as MKOA does not mean you are "healthy", and again, what does "healthy" mean? Authors should use more clinically based terminology that is more objective in nature for defining the controls. Again, within the methods, inclusion and exclusion criteria for the controls should be added.

Under gait analysis, it is discussed in lines 108-109 that sensors were attached via Velcro bands, but how were the bands adjusted so that it minimized movement and was this standardized between patients? Include this information. Also, lines 109-110 discuss velocity was self-selected, which is highly unusual since velocity can impact gait mechanics according to numerous studies, thus, include a statement as to why self-selected velocity was utilized for this study. Was this to replicate what may be done in a clinical setting since standardizing velocity is difficult to do outside of a research setting? Also, include within this same area where the data collection was done including the type of footing. Line 112 mentions only three gait tests. Why were only three selected? Is there other studies that indicated this would be a preferred number of gait tests for each subject and were there any acclimation trials before the three tests to allow the subject to get use to the sensors, particularly the Velcro bands? If not, why and could that impact the first one or two tests as they got use to the sensors? If there was an acclimation period, describe the acclimation period and how it was ensured that the subjects did not become fatigued throughout this acclimation period prior to the three tests. Also give the timing of when and how long this acclimation period was. In addition, what was the timing of the three tests in that was there a break between tests, and if so, how long of a break and what were participants able to do between these breaks.

As for minor corrections within the methods section, enlarge figure 1 as it is difficult to see, maybe dividing into two figures. Lines 126-127 is an incomplete sentence. Lines 128-133 repeat lines 110-120, thus, remove accordingly. Enlarge the words at the top of figure 2. 

As for the results, again, reword "healthy" from "healthy knee" to refer to the controls. Do define in the methods what made these individuals fall within the control section in that was it just a lack of MKOA or other aspects of clinically relevant knee conditions that were utilized as exclusion criteria? Why only 24 in the control compared to 69 MKOA patients? Were the statistical analysis the best suited for an unbalanced data set? What limited the authors from having a balanced data set? Seems odd to have a control sample size that is less than half of the treatment group, thus, explain. Table 4 is hard to read with the two tables put side by side, and thus, it may be helpful to have them set top to bottom where authors can increase font size within each table . For figure 4 the x-axis font is difficult to read. Again, instead of two figures side by side, authors may want to put one figure on top of the other. Authors need to add more explanation into the figure title to explain the figures more as to what the reader is looking at and authors should add additional colors to the figures to help to distinguish different grades and groups. 

As for the discussion, within line 236 the authors mention "early phase", but how was that determined for the patients used in this study? Without previous data collection followed by later collection, well after the initial trials, it is hard to signify that this was an early phase to the MKOA for these patients. Authors are encouraged not to overreach in what was found within this study. In line 240, the authors mention "our previous study", which is confusing as to whether that was information presented within this study or not. Authors should just cite that study similar to other studies referenced within the manuscript. Again, in lines 257 and 258, the word "healthy" is utilized, but without clearly defining what is considered "healthy", this use of vague, subjective terminology can be confusing. The authors are encouraged to look at different, more objective, clinically based terminology. A person without MKOA does not mean they are healthy as they can have many underlining health conditions including ones associated with the knee that would exclude them from the consideration of being healthy. Similarly, the term "normal" is utilized in lines 259, 268, and 294, but what is "normal". Lines 259 and 268 are discussing the gait as it relates to "normal", but the authors first don't define what the walk is and then what is considered a "normal" walk. This needs to be done in a clear objective manner that includes indicators of what would be considered abnormal. Unfortunately, a normal, sound gait often includes a prescribed velocity, and thus, authors will have to adjust accordingly. Line 294 refers "normal" as related to the knee, but again, the authors must first define what is considered a normal knee. Authors may want to consider more objective, clinically based terminology besides "healthy" and "normal". Lines 282-284 and 286-287 are both incomplete sentences. In line 300, instead of starting the sentence "The Further study", authors should say "Further studies".  Again, there are additional issues with grammar within the manuscript, and thus, authors should take additional time within their revisions to address this problem as the material given within the manuscript will be better appreciated once that aspect of the manuscript is corrected. 

Author Response

Thank you for your review.

Reviewer 2 Report

This is a very interesting article. However, there are some questions to be clarified before publication.

1.     In this abstract, the author provided a qualitative description of the analysis results, but no quantitative results. It would be better if these results could be supplemented.

2.     Please provide full name of the abbreviation of “BMI” in the line of 99.

3.     Whether “ethics review” has been approved since some human beings have participated in this study as stated between line 98 to line 99.

4.     The fonts in Figure 1 are too small and does not match the article.

5.     The author simplified the analysis process by citing their previous publication in Line 122, however, it would be more convenient to readers if they provided the “key analysis framework”.

6.     The author wrote “no clear candidate has been proposed to replace the conventional silicon MOSFETs.” Line 31 to line 32.  If the author means that the main purpose of the TFT is to replace the MOSFETS, I would say it was not a suitable thought. This expression caused a misleading to readers.

7.     The “statistical analysis” part need some pictures to give more detail description of the analysis process.

8.     All abbreviations in the text should have a full description, such as: KL, HSD or something like that.

9.     The fonts of all tables in the text are too small to the whole article.

10.   Please re-arrange the expression Line 60 to Line 62

11.   The conclusion part seems to be too little.

Author Response

Thank you for your review.
